# Contribution of Multiple Inherited Variants to Autism Spectrum Disorder (ASD) in a Family with 3 Affected Siblings

**DOI:** 10.3390/genes12071053

**Published:** 2021-07-08

**Authors:** Jasleen Dhaliwal, Ying Qiao, Kristina Calli, Sally Martell, Simone Race, Chieko Chijiwa, Armansa Glodjo, Steven Jones, Evica Rajcan-Separovic, Stephen W. Scherer, Suzanne Lewis

**Affiliations:** 1Department of Medical Genetics, University of British Columbia (UBC), Vancouver, BC V6H 3N1, Canada; jasleen.dhaliwal@bcchr.ca (J.D.); yqiao@mail.ubc.ca (Y.Q.); kcalli@mail.ubc.ca (K.C.); sally.martell@ubc.ca (S.M.); cchijiwa@cw.bc.ca (C.C.); sjones@bcgsc.ca (S.J.); 2BC Children’s Hospital, Vancouver, BC V5Z 4H4, Canada; evica@mail.ubc.ca; 3Department of Pediatrics, University of British Columbia (UBC), Vancouver, BC V6T 1Z7, Canada; srace@bcchr.ca (S.R.); aglodjo@cw.bc.ca (A.G.); 4The Centre for Applied Genomics and McLaughlin Centre, Hospital for Sick Children and University of Toronto, Toronto, ON M5G 0A4, Canada; Stephen.Scherer@sickkids.ca; 5BC Children’s and Women’s Health Center, Vancouver, BC V6H 3N1, Canada; 6Michael Smith Genome Sciences Centre, Vancouver, BC V5Z 4S6, Canada; 7Department of Pathology and Laboratory Medicine, University of British Columbia (UBC), Vancouver, BC V6T 1Z7, Canada

**Keywords:** Autism Spectrum Disorder (ASD), whole-genome sequencing (WGS), gene variants, *RELN* gene, *SHANK2* gene, quantitative trait hypothesis, complex genetics

## Abstract

Autism Spectrum Disorder (ASD) is the most common neurodevelopmental disorder in children and shows high heritability. However, how inherited variants contribute to ASD in multiplex families remains unclear. Using whole-genome sequencing (WGS) in a family with three affected children, we identified multiple inherited DNA variants in ASD-associated genes and pathways (*RELN*, *SHANK2*, *DLG1*, *SCN10A*, *KMT2C* and *ASH1L*). All are shared among the three children, except *ASH1L*, which is only present in the most severely affected child. The compound heterozygous variants in *RELN,* and the maternally inherited variant in *SHANK2,* are considered to be major risk factors for ASD in this family. Both genes are involved in neuron activities, including synaptic functions and the GABAergic neurotransmission system, which are highly associated with ASD pathogenesis. *DLG1* is also involved in synapse functions, and *KMT2C* and *ASH1L* are involved in chromatin organization. Our data suggest that multiple inherited rare variants, each with a subthreshold and/or variable effect, may converge to certain pathways and contribute quantitatively and additively, or alternatively act via a 2nd-hit or multiple-hits to render pathogenicity of ASD in this family. Additionally, this multiple-hits model further supports the quantitative trait hypothesis of a complex genetic, multifactorial etiology for the development of ASDs.

## 1. Introduction

According to the National Autism Spectrum Disorder Surveillance System (NASS) (2018), Autism Spectrum Disorder (ASD) is diagnosed in 1 in 66 Canadian children and youth (ages 5–17), making it one of the most common neurodevelopmental disorders in children. Almost half of the cases have unknown causes. Although both environmental and genetic factors are known to be involved in ASD pathogenesis, the genetic heritability of autism can reach up to 90% in early twin studies [1], indicating a strong genetic influence on ASD. To date, hundreds of genes and DNA copy number variant (CNV) loci have been reported to be associated with ASD susceptibility [2]. However, even the most common single-nucleotide variant (SNV) or genomic CNV micro-deletion or -duplication account for no more than 3% of ASD cases [3], suggesting a highly heterogeneous genetic background. Whole-genome sequencing (WGS), as a state-of-the-art high-throughput technology, has improved ASD diagnosis by 20% and has shown the potential to become a first-tier genetic test for neurodevelopmental disorders [4,5]. Using WGS and genome-wide chromosomal microarray, it has now been demonstrated that de novo mutations can be found in 5–30% of ASDs, especially in simplex families with sporadic cases [6]. However, the genetics of inherited rare variants are poorly understood, despite the fact that they can be found in 3–5% of cases of ASD [3]. Therefore, the cause of ASD in multiplex (MPX) families, in which more than one member is affected, remains largely unknown.

As part of a collaboration established between the iTARGET Autism project (http://www.itargetautism.ca/) and the Autism Speaks MSSNG project (https://www.mss.ng/), we used WGS to detect SNVs, small insertions and deletions (indels), and genomic CNVs in an MPX family with three affected children, aiming to investigate disease-causing/disposing variants which are segregated with the phenotype in this family.

## 2. Materials and Methods

### 2.1. Whole Genome Sequencing (WGS)

WGS was performed using the following platforms (the Illumina HiSeq X WGS platform for Sib-2 and Sib-3; Complete Genome on Sib-1) at the Toronto Sick Kids Hospital through our collaboration. The data were aligned with the reference genome (GRCh38). Both vcf and bam files were imported into a commercial software VarSeq (GoldenHelix, Inc., Bozeman, MT, USA) for SNVs/Indels and CNVs analysis. In brief, CNVs were generated by the Binned Region Coverage (minimum 10 Kb) and CNV algorithm in VarSeq. The SNVs/Indels were filtered by quality control (QC), annotated by over 20 databases in VarSeq, and interpreted by our internal pipeline. Our major criteria include QC for Read Depth ≥ 10, Genotype Quality ≥ 20); MAF ≤ 0.05 in gnomAD [7] for homozygous recessive and compound heterozygous variants; MAF ≤ 0.01 for de novo, X-linked, ASD candidate genes, imprinting genes, incidental findings (59 genes on the ACMG incidental finding list), loss-of-function (LOF) variants, and others on our in-house gene lists. Our criteria on variants with a disease-causing effect include: (1) Missense (missense, inframe-deletion/insertion, 5_prime_UTR_premature_start_codon) and LOF (stop-gain, stop-loss, frameshift, essential splice, and initiator codon) variants. (2) Frequency in gnomAD meets the criteria for different inheritance patterns, described above. (3) The PHRED score of CADD [8] is >20 on VarSeq software. (4) At least 2 out of 5 bioinformatics tools were predicted as damaging according to the algorithm on VarSeq software (SIFTPred, Polyphen2HumVarPred, MutationTasterPred, MutationAssessorPred, FATHMMPred). (5) Genes listed on SFARI (https://gene.sfari.org/, accessed on 1 October 2020). (6) Other evidence from OMIM (https://omim.org, accessed on 1 October 2020), HGMD (Professional 2021.1), ClinGen (https://clinicalgenome.org/, accessed on 1 October 2020), Genereviews (https://www.ncbi.nlm.nih.gov/books/NBK1116/, accessed on 1 October 2020), ClinVar (https://www.ncbi.nlm.nih.gov/clinvar/, accessed on 1 October 2020), PubMed, GeneCards (https://www.genecards.org/, accessed on 1 October 2020), AutDB (http://autism.mindspec.org/autdb/Welcome.do, accessed on 1 October 2020), Genatlas (http://genatlas.medecine.univ-paris5.fr/, accessed on 01 October 2020), Locus Reference Genomic (LRG) (https://www.lrg-sequence.org/, accessed on 1 October 2020), Decipher (https://www.deciphergenomics.org/, accessed on 1 October 2020), etc. (7) Correlation analysis with phenotypes collected from patient’s charts.

### 2.2. Subjects

The participating family for this study has three boys with ASD, born to phenotypically normal, non-consanguineous parents. The mother is a β-thalassemia carrier. The mother’s paternal aunt and uncle both exhibit symptoms of obsessive-compulsive disorder (OCD) and anxiety disorder (unconfirmed). The mother’s father is anemic and is strongly suspected to have high-functioning ASD. The father is an α-thalassemia carrier, and the father’s nephew has Pervasive Developmental Disorder—Not Otherwise Specified (PDD-NOS). All of the boys were diagnosed with ASD using gold-standard Autism Diagnostic Observation Schedule—Generic (ADOS-G) and Autism Diagnostic Interview—Revised (ADI-R) psychometric measures. Sib-1 (15 years old) has Attention Deficit Hyperactivity Disorder (ADHD), anemia, decreased pain sensitivity, and no outwardly syndromic features. Sib-2 (14 years old) has slightly coarse facies. He shows the most prominent cognitive deficits and a greater severity of ASD. He has a history of dysphagia and no anemia (not a thalassemia carrier). Sib-3 (8 years old) has suspected IUGR, anemia since birth, astigmatism, and dysphagia with problematic swallowing. None of the affected siblings show any outwardly syndromic or dysmorphic features.

## 3. Results

### 3.1. Single Nucleotide Variant (SNV) Findings

No rare de novo variants with significant disease-causing effects were found in this family. Instead, we identified multiple rare inherited variants in ASD candidate genes, mostly shared by all three siblings, which we assert collectively contribute to the pathogenesis of ASD in these three siblings (Figure 1 and Table 1).

First, all three affected children were found to share rare compound heterozygous variants in *RELN* (NP_005036.2: p.Ser630Arg and p.Val1153Ile). These variants are located in exons 15 and 25 within repeat 1 and 2 of the Reelin protein domain, respectively. Both variant loci are highly conserved in many species. The missense p.Ser630Arg variant was inherited from the mother, present in dbSNP with <3% in normal population databases, including gnomAD, 1000 Genome, and NHLBI ESP6500. Multiple bioinformatics tools predict this variant as damaging (SIFT, MutationTaster, Polyphen2 HDIV, LRT, FATHMM MKL) with the PHRED score of CADD as 25. The missense p.Val115Ile variant was paternally inherited, absent from dbSNP or ClinVar, with <0.05% in normal population databases and a PHRED score of CADD as 19. Meanwhile, we also detected a paternally inherited missense coding single nucleotide polymorphism (SNP) mutation of *RELN* (rs36269) in dbSNP in exon 22 (NP_005036.2: p.Leu997Val). All three siblings share this SNP. Contradictory results from the literature suggest that this variant either significantly contributes to the susceptibility of ASD [9,10,11], or does not [12,13,14].

The second possibly disease-associated, rare variant found to be shared by all three affected boys was a maternal missense variant in *SHANK2* (NP_573573.2:p.Pro1184Ser). This missense variant is absent from dbSNP, ClinVar, 1000 Genome, and NHLBI ESP6500 with <0.001% in gnomAD. The variant in our proband occurred at a location 10 amino acids away from the conserved SAM domain of Shank family proteins. It was predicted to be damaging by multiple bioinformatics tools (SIFT, MutationTaster, Polyphen2 HDIV, PROVEAN, FATHMM MKL) with the PHRED score of CADD as 24.

In addition to SNVs in *RELN* and *SHANK2*, several other inherited variants were also found in the three affected siblings, which involve ASD candidate genes and pathways. These include a paternal variant in *KMT2C* (NP_733751.2:P.Ser4300Pro), a maternal variant in *DLG1* (NP_004078.2:P.Ala295Val), and compound heterozygous variants in *SCN10A* (NP_006505.3:P.Arg1142Pro and P.Thr1181Met). A maternal variant in *ASH1L* (NP_060959.2:p.Gln433Pro) was identified only in Sib-2. They are all rare missense variants with high CADD scores (20–34).

### 3.2. Copy Number Variants (CNVs) Findings

Using genome-wide chromosomal microarray (Affymetrix CytoScan HD platform; Affymetrix Inc., Santa Clara, CA, USA), we did not find any abnormal CNVs in Sib-2 or Sib-3. However, we identified an 18 kb deletion in Sib-1: arr[hg38]16p13.3(166421-184365)x1, involving *HBA1*, *HBA2*, *HBQ1*, and *HBM* (α hemoglobin locus). CNV analysis from WGS confirmed that this deletion was inherited from the father, who is an α-thalassemia carrier. Consistent with the family history of thalassemia (mother is a β-thalassemia carrier), a maternal pathogenic LOF mutation in HBB (c.126_129delCTTT, p.Phe42Leufs) was identified in Sib-3. Both Sib-1 and Sib-3 have anemia, while Sib-2, without anemia, does not have either of these two mutations.

## 4. Discussion

Using WGS, we identified multiple rare inherited variants in a multiplex family with three affected children. The major ASD risk factors include compound heterozygous variants in *RELN* and a maternal missense variant in *SHANK2*, which are shared in all three children. *RELN* encodes Reelin, a large glycoprotein secreted by GABAergic interneurons and glutamatergic cerebellar neurons. It plays a vital role in Purkinje cell positioning during brain development and in modulating adult synapse transmission and plasticity [15,16]. A deficiency in Reelin signaling and pathologic impairment of Reelin secretion was found to contribute to ASD risk [17]. Reeler mice, which lack the *Reln* gene, show impaired GABAergic Purkinje neuron expression/positioning during cerebellar development [18], and exhibit abnormal social and repetitive behaviors similar to ASD behaviors [19]. *RELN* is a high-confidence ASD-associated gene with many rare de novo and inherited missense variants identified in patients with ASD [20]. However, most of these variants lack functional analysis, and some of them are inherited from unaffected parents, suggesting that the *RELN* mutation shows variable expression or incomplete penetrance; hence, it is unable to cause ASD by itself [2,20,21,22]. Instead, *RELN* variants may act as risk factors to co-act and synergize with other genetic or environmental factors, collectively contributing to an ASD phenotype. Recently, Sanchez-Sanchez et al. identified rare compound heterozygous missense variants in *RELN* in a patient with ASD [23]. Using iPSC-derived neural progenitor cells (NPCs) from their patient, they provided experimental evidence that the identified variants are deleterious, and lead to diminished Reelin secretion and impaired Reelin–DAB1 and mTORC1 signal pathways. Moreover, they found a de novo splice-site variant in the *CACNA1H* gene and an inherited missense variant in the *CYFIP1* gene, both of which are connected with mTORC1 and Reelin–DAB1 signaling cascades. The findings of Sanchez-Sanchez et al. support our hypothesis and results for a multiple-hits or quantitative risk variant model of ASD etiology, and confirm that heterozygous or compound heterozygous recessive variants in the *RELN* gene are cannot cause ASD by themselves.

*SHANK2* is a post-synaptic scaffolding gene located at the post-synaptic density of glutamatergic synapses. It plays a crucial role in the excitatory synaptic transmission [24,25], and the formation of dendritic spines [26,27]. *Shank2* knockout mouse models show defects in excitatory synapse function and display an ASD-like phenotype, including abnormalities in motor behavior, vocalization, and socialization [25,28]. In humans, inherited variants in *SHANK2* were found to be shared by multiple affected siblings and their slightly affected or unaffected parents [26,29,30], suggesting that additional genetic/epigenetic factors, together with inherited *SHANK2* mutations, might be necessary to develop ASD [30,31].

Interestingly, both *RELN* and *SHANK2* are associated with neuron activities, including synaptic function and GABAergic interneuron signaling pathways. An imbalance in these pathways has been hypothesized as an underlying mechanism of ASD (for review, see [32]). *RELN*-involved Reelin-signaling pathway interacts with neuroligins indirectly through a scaffolding protein, PSD-95, which is anchored to the cytoskeletons through SHANK proteins [21,33]. However, the co-existence of variants in both *RELN* and *SHANK2* genes found in subjects with ASD has not previously been reported.

Additionally, we identified several other rare inherited variants in ASD associated genes and pathways among the affected children, including *DLG1*, *SCN10A*, *KMT2C*, and *ASH1L*. *DLG1* gene encodes a scaffolding protein that plays a vital role in normal development, including synaptogenesis [34]. It is hypothesized that alterations in scaffolding protein dynamics could be part of the pathophysiology of ASD [35]. *SCN10A* gene is a part of the sodium channel family and is involved in modulating the activity of neurons [36]. A limited number of rare missense variants in *DLG1* and *SCN10A* have been reported in cases with ASDs [35,37]. Whether or not they are novel ASD candidate genes needs further functional analysis. Both *KMT2C* and *ASH1L* are histone methyltransferases genes with shared domains (an AT hook DNA-binding domain and a PHD-finger motif) and functions. They are important histone regulator genes and involved in chromatin organization [38,39]. Disruption of histone methylation has been reported in neurodevelopmental disorders and ASDs [20]. Variants in these two strong ASD candidate genes are also widely reported in ASD cases [2,20,40].

All of the above detected familial variants were found to be shared by all three children, except for the variant in *ASH1L* that was only detected in Sib-2. Sib-2 is the most severely affected among the three children. It has been demonstrated that patients with two or more de novo mutations in ASD candidate genes showed more severe phenotypes [41,42]. Whether this inherited *ASH1L* missense variant may contribute to the more severe ASD phenotype in Sib-2 needs further functional studies.

Noticeably, *RELN*, *SHANK2*, and *DLG1* are all involved in synapse functions, while *KMT2C* and *ASH1L* share chromatin organization in function. The regulation and maintenance of synapse activity and chromatin organization are closely associated with ASD pathogenesis [2,20,43]. Our data suggest that the convergence of the variants of these genes in certain ASD relevant pathways might contribute to the risk of ASD. Individually, each of the variants that we identified in this family, was found not to play a significant role in causing ASD. However, their co-occurrence and co-segregation amongst the three affected children, especially their interconnected gene functions and mechanistic pathways, might indicate their involvement in the ASD pathogenesis in this family. For example, they may co-act and synergize together to increase the genetic mutation load resulting in the disruption of one or multiple gene signal pathways. In addition, these data further support that there is advantage to using WGS in genotype-phenotype and pathway-based analysis of genomic data, which is essential to genetic counseling and management decisions based on ASD genomic etiology, rather than symptoms alone.

One of the limitations of our study is the lack of functional testing of these variants. The pathogenicity of these identified variants isbased solely on in silico prediction tools, although they are widely used in the current sequencing analysis. Additionally, more evidence is required for future family-based WGS analysis in more ASD multiplex families.

## 5. Conclusions

We identified multiple inherited missense variants of the six ASD-related genes and pathways shared in this MPX family. Our data suggest that each of the variants may have a variable subthreshold or subtle clinical impact. However, together they may collectively and quantitatively contribute to an additive “tipping over the ASD threshold” effect, supporting a multiple-gene hit model for the complex genetic basis and development of ASD.

## Figures and Tables

**Figure 1 genes-12-01053-f001:**
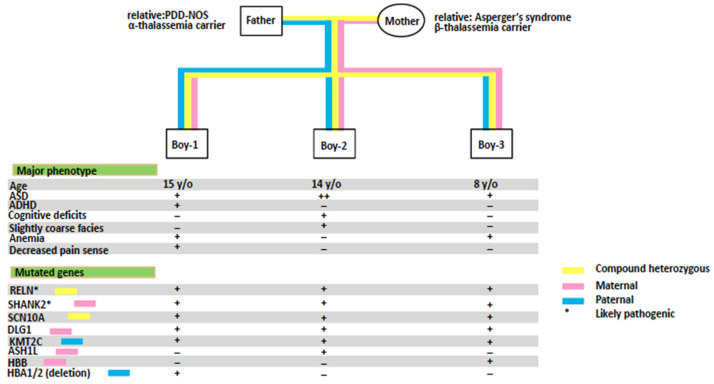
Clinical features and genomic findings distributed in the multiplex family.

**Table 1 genes-12-01053-t001:** ASD-related genes and variants identified in the multiplex family.

Gene	Variant	Inheritance	Frequency	CADD PHRED Score	Gene Function
***RELN***	NM_005045.3:c.3457G>A;NP_005036.2:p.Val1153Ile;/NM_005045.3:c.1888A>C;NP_005036.2:p.Ser630Arg	Compound heterozygous (in all 3 boys)	<0.0004/<0.026	19/25	Synaptic function and neuronal migration
***SHANK***	NM_133266.4:c.3550C>T;NP_573573.2:p.Pro1184Ser	Mat (in all 3 boys)	≤8.236 × 10^−6^	24	Synapse formation, maturation and structural plasticity
***KMT2C***	NM_170606.2:c.12898T>C;NP_733751.2:p.Ser4300Pro	Pat (in all 3 boys)	<0.0004	24	Leukemogenesis and chromatin organization
***DLG1***	NM_004087.2:c.884C>T; NP_004078.2:p.Ala295Val	Mat (in all 3 boys)	<0.0016	33	Synapse formation and function
***SCN10A***	NM_006514.3:c.3425G>C;NP_006505.3:p.Arg1142Pro/NM_006514.3:c.3542C>T;NP_006505.3:p.Thr1181Met	Compound heterozygous (in all 3 boys)	NR/<0.0022	34/14	Sodium channel, modulating the activity of CNS neurons
***ASH1L***	NM_018489.2:c.1298A>C;NP_060959.2:p.Gln433Pro	Mat (only in Boy-2)	≤0.00003	20	Chromatin remodeling and organization

Note: Mat: maternal. Pat: paternal.

## Data Availability

The data presented in this study are available in the manuscript, or can be obtained from the authors upon written request to the corresponding author.

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
