# Peer review of "Contribution of Multiple Inherited Variants to Autism Spectrum Disorder (ASD) in a Family with 3 Affected Siblings"

_genes, 2021, doi:10.3390/genes12071053_

Round 1

Reviewer 1 Report

The paper “Contribution of multiple inherited variants to Autism Spectrum Disorder (ASD) in a family with 3 affected siblings” is a valuable case report describing the results of whole genome sequencing (WGS) in one multiplex ASD family. This report is particularly valuable in light of the fact, highlighted in the manuscript, that the genetics of inherited ASD is poorly understood. This is an important contribution to the literature with some issues that should be addressed prior to publication as outlined below.

1.    Please describe the specific criteria used to determine that there are “no rare de novo variants with significant disease-causing effect”. How was “disease-causing effect” defined? 

2.    Similarly, and importantly, WGS identifies a large number of missense variants in any given individual, including many that are present in the Simon’s SFARI database. The claim that all such variants identified in the three affected offspring in this family are causative requires some additional supporting evidence. It may be the case that this is true and the variants in interesting genes are acting together, but this has not been established. It is important and worthwhile to report specific variants detected, and the strength of this report is that multiple variants segregate with the phenotype in this family. It would cleave closer to the evidence and the actual data to highlight this fact and make a less strong claim about the genetic etiology of the disease, based solely on findings from one family. This paper presents evidence in one family, opening an avenue that can be followed up by further study in additional families. This is a valuable contribution and should be reported but the conclusions should not overreach the evidence.

3.    In addition, the pathogenicity of the identified variants has not been established. No functional data is presented to support that these variants will impact the encoded protein in any way. In silico predictions are often useful for variant filtration and prioritization purposes, but do not on their own constitute evidence of variant pathogenicity.

Reviewer 2 Report

This is a nice little study on the multiplex family of 3 brothers with ASD demonstrating polygenic inheritance. The manuscript is well written. It is timely in that polygenic inheritance is now being shown to be common in ASD, and this article shows what can be identified by genetic testing in 2021.

I do suggest referencing some additional papers suggesting that the de novo rate in ASD might be much higher than the low numbers quoted. Both polygenic and de novo mechanisms are common in ASD.

Did the genetic data lead to any change in clinical management, perhaps aimed at GABA or sodium channel pathways?
